# Validation of a Computerized Adaptive Test Suicide Scale (CAT-SS) among United States Military Veterans

Lisa A. Brenner[1,2,3,4]⊕*, Lisa M. Betthauser[1,2]⊕, Molly Penzenik[1]⊕, Nazanin Bahraini[1,2,3]⊕, Robert D. Gibbons[5,6,7]⊕

1 Veterans Health Administration Rocky Mountain Mental Illness Research Education and Clinical Center (RM MIRECC), Rocky Mountain Regional Veterans Affairs Medical Center, Eastern Colorado Health Care System, Aurora, Colorado, United States of America, 2 Department of Physical Medicine & Rehabilitation, University of Colorado, Anschutz Medical Campus, Aurora, Colorado, United States of America, 3 Department of Psychiatry, University of Colorado, Anschutz Medical Campus, Aurora, Colorado, United States of America, 4 Department of Psychiatry & Neurology, University of Colorado, Anschutz Medical Campus, Aurora, Colorado, United States of America, 5 Department of Medicine, University of Chicago, Chicago, Illinois, United States of America, 6 Center for Health Statistics and Departments of Public Health Sciences (Biostatistics), Psychiatry, Comparative Human Development, Chicago, Illinois, United States of America, 7 Committee on Quantitative Methods, University of Chicago, Chicago, Illinois, United States of America

⊕ These authors contributed equally to this work.
* Lisa.2.Brenner@cuanschutz.edu

**Data Availability Statement:** The full dataset cannot be made publicly available due to privacy concerns and restrictions imposed by the Colorado Multiple Institutional Review Board. All relevant de-

## Abstract

To validate the Computerized Adaptive Test Suicide Scale (CAT-SS), Veterans completed measures at baseline (n = 305), and 6- (n = 249), and 12-months (n = 185), including the CAT-SS (median items 11, duration of administration 107 seconds) and the Columbia-Suicide Severity Rating Scale (C-SSRS). Logistic regression was used to relate CAT-SS scores (baseline) to C-SSRS assessed outcomes (active ideation with plan and intent; attempt; interrupted, aborted or self-interrupted attempt, or preparatory acts or behaviors; all outcomes combined). A mixed-effects logistic regression model was used to evaluate the relationship between the lagged CAT-SS scores and outcomes (6- and 12-months). The baseline CAT-SS demonstrated predictive accuracy for all outcomes at 6-months, and similar results were found for baseline and all outcomes at and through 12-months. Longitudinal analysis revealed for every 10-point change in the CAT-SS there was a 50–77% increase in the likelihood of suicide-related outcomes. The CAT-SS demonstrated added value when compared to current suicide risk prediction practices.

## Introduction

In the United States, rates of suicide have been increasing among military and civilian cohorts [1, 2]. According to work by Ahmedani et al. [3], almost 30% of individuals who died by suicide had a healthcare visit in the week prior to their death. Recognizing the importance of risk screening within healthcare systems, in 2016 the Joint Commission released a Sentinel Event

identified data are included in the manuscript. For investigators with appropriate authorizations within the Department of Veterans Affairs, requests for data access can be made to VHAECHMIRECCAdmin@va.gov.

**Funding:** Funding was provided by the Veterans Health Administration, Office of Mental Health and Suicide Prevention; NIMH Grant #RO1 MH100155-06. The Office of Mental Health and Suicide Prevention did not influence the decision to submit this manuscript for submission.

**Competing interests:** The views, opinions, and/or findings contained in this article are those of the author(s) and should not be construed as an official Department of Veterans Affairs position, policy, or decision unless so designated by other documentation. Dr. Brenner has received royalties from American Psychological Association Publishing and Oxford University Press. Dr. Gibbons has been an expert witness for the US Department of Justice, Merck, Glaxo-Smith-Kline, Pfizer and Wyeth and is a founder of Adaptive Testing Technologies, which distributes the CAT-MH™ battery of adaptive tests in which the CAT-SS is included. The terms of this arrangement have been reviewed and approved by the University of Chicago in accordance with its conflict of interest policies.

Alert, which recommended that universal suicide risk screening be implemented [4]. Ideally, such efforts would facilitate identification of those with occult risk (individuals who may disclose suicidal thoughts and behaviors only if they are directly asked) who may not be engaged in mental health treatment [5]. Nonetheless, options and evidence regarding tools which can be used to facilitate universal risk screening remain limited [4].

As screening for depression frequently occurs in primary care settings, often using the Patient Health Questionniare-9 (PHQ-9) [6], efforts to evaluate the utility of the PHQ-9 as a suicide risk screener have been undertaken. However, likely related to the measure only containing one item (item 9) specifically focused on suicidal ideation ("bothered by thoughts of being dead or of hurting yourself in some way"), as well as the reality that a sizable number of individuals' risk for suicide is related to factors other than depression (e.g., chronic pain, anxiety), results have been mixed. In specific, data regarding psychometric properties (e.g., positive predictive value) have been less than ideal [6–8]. Moreover, results from most rapid screeners like the PHQ-9 item 9 [6] often do not provide the clinician with information regarding risk severity or magnitude [7].

In addition, many suicide risk screening measures (e.g., the Columbia Suicide Rating Scale (C-SSRS)-Screener) [9] include items solely focused on suicidal ideation and behavior; thereby limiting the ability to measure "the full spectrum of suicidal symptomatology" [7; pp. 1376]. Ideally, suicide risk screening approaches would incorporate personalized items associated with a range of risk factors. Tailoring screening measures while maintaining psychometric properties requires implementation of novel approaches such as computerized adaptive testing (CAT) based on unidimensional or multidimensional item response theory (M/IRT). Traditional mental health measures are based on classical test theory, where all respondents receive all items and which are equally weighted in terms of deriving the test score, which is the often the summation of the individual item scores, rated either dichotomously or as polytomous Likert scale items. In contrast, IRT-based CAT uses unidimensional or multidimensional item response theory (MIRT) to pre-calibrate a large "bank" of symptom-items, that are then adaptively selected to match the severity of the person's disorder, which is adaptively estimated from the responses to prior items administered [7; pp. 1377]. As a result, different items are administered to different respondents, targeted to their level of severity on the underlying construct of interest (in our case suicide risk). For further information regarding MIRT-based CAT see Gibbons et al., 2008 [10] and Gibbons et al., 2016 [11].

Thus, Gibbons and colleagues developed and conducted an initial validation study on the Computerized Adaptive Test-Suicide Scale (CAT-SS) [7]. Using data from individuals receiving outpatient psychiatric treatment, the team was able to calibrate the CAT-SS, and demonstrate that the CAT-SS measured suicide risk severity using a mean of 10 items, in under two minutes. Moreover, initial validity was demonstrated comparing CAT-SS and C-SSRS structured clinical interview results among those seeking care in two non-Veterans Affairs emergency departments (University of Chicago and University of Massachusetts). Contrasting the CAT-SS high-risk group to the no-risk group a sensitivity of 1.0 and specificity of 0.92 were found for the C-SSRS active ideation category. Per the authors, additional prospective validation efforts, including prediction of future suicidal events, were warranted. Towards this end, members of this team conducted a longitudinal study among Veterans eligible for Veterans Health Administration (VHA) care to validate the CAT-SS self-report measure in terms of its ability to predict future suicide events based on repeated C-SSRS clinical interviews at 6-months and 12-months following the baseline CAT-SS assessment.

## Methods

### Participants

This study was conducted according to the guidelines laid down in the Declaration of Helsinki and all procedures involving human participants were approved by the Colorado Multiple Institutional Review Board (COMIRB). Participants (n = 305) were recruited from a mountain state metropolitan VA health care system between April 2017 and February 2019. Recruitment strategies included posting flyers at local facilities, contacting Veterans who had participated in previous research or who indicated interest in participating in research, and encouraging providers to tell patients about the study. Veterans were eligible if they were between the ages of 18 and 89 and able to provide written informed consent, which was obtained. The number of veterans who completed measures at each timepoint is as follows: baseline, n = 305; 6-month follow-up, n = 249; 12-month follow-up, n = 185.

### Measures

Computerized Adaptive Test-Suicide Scale (CAT-SS) [7]* is an adaptive measure, comprised of 111-items, which dimensionally measures suicide risk severity on a 100-point scale with 5 points of precision. The scores are also thresholded to yield categories of low, moderate, and high risk.

The Columbia-Suicide Severity Rating Scale (C-SSRS) [9]* is a clinician-administered inter-view used to evaluate suicidal ideation (including intensity) and suicide-related behavior (e.g., preparatory, attempt).

Structured Clinical Interview for DSM-5 Disorders (SCID-5) Research Version [12] is a reliable and valid semi-structured interview used to diagnose Axis I psychiatric disorders in clinical and research settings. The SCID-5 was used to determine current presence of the following disorders: Bipolar I and II; Major Depressive; Alcohol Use; Substance Use, Generalized Anxiety; and, Sleep. The trauma/PTSD L Module of the Structured Clinical Interview for SCID-5 [12] was used to assess Criterion A events. If a Criterion A event and at least one current symptom was endorsed, the Clinician-Administered PTSD Scale for DSM-5 CAPS-5 was administered. The CAPS-5 is the gold standard for assessing PTSD, and was used to determine current PTSD diagnosis [13].

Rocky Mountain MIRECC Demographic Questionnaire was used to gather information on topics such as participant age, gender, race/ethnicity, education, period of military service, and combat exposure.

*Measures administered at baseline, and 6- and 12-month follow-up appointments.

### Procedures

Data were collected at three timepoints (baseline, 6- and 12-month follow-up). After confirming eligibility, Veterans were invited to an in-person baseline study visit. Informed consent was obtained prior to administration of clinical interviews listed above, self-report measures (not included in this study), and the CAT-SS. Study team members were clinically trained to administer the measures and interview schedules were reviewed by licensed clinicians.

To facilitate retention, participants were re-contacted at approximately 6 months post the baseline study visit and offered an in-person or telephone visit. During this visit, the CAT-SS was re-administered. In addition, reminder letters to invite completion of the 12-month follow-up were sent 1–3 months prior to their 12-month window to promote retention. The final in-person study visit was conducted approximately 12 months following the baseline assessment, and the CAT-SS was again re-administered. Participants were compensated for all study

visits. Two Veterans who had incomplete data at the 6-month visit and one at the 12month visit were removed from analyses. Reasons for attrition were not collected, however, Veterans were invited to complete the 12-month follow-up regardless of their completion of their 6-month follow-up. The final sample size for analysis was n = 265.

## Statistical analyses

Logistic regression was used to relate the CAT-SS scores at baseline to the C-SSRS assessed outcomes (active ideation with plan and intent; attempt; interrupted, aborted or self-inter-rupted attempt, or preparatory acts or behaviors; all outcomes combined) at 6 months, and the CAT-SS scores at baseline and 6-months to the outcomes at 12-months, and all events between baseline and 12-months. From the logistic regression model, we generated a receiver operating characteristic (ROC) curve and computed the area under the ROC curve (AUC). We also examined the unique contribution of the CAT-SS in predicting suicide-related outcomes over and above what has traditionally been considered a robust predictor, a suicide attempt within the past year. To test this, logistic regression models with: (a) previous suicide attempt in the past year; (b) the CAT-SS; and, (c) previous suicide attempt in the past year and the CAT-SS were fitted to these data and the AUCs statistically compared.

To study longitudinal trends in C-SSRS assessed outcomes (active ideation with plan and intent; attempt; interrupted, aborted or self-interrupted attempt, or preparatory acts or behav-iors; all outcomes combined), a mixed-effects logistic regression model was used to perform a longitudinal analysis of the relationship between the lagged CAT-SS scores and suicide-related outcomes at 6 and 12 months (i.e., CAT-SS score at baseline predicting suicide-related out-comes at month 6 and CAT-SS at month 6 predicting suicide events at 12 months). CAT-SS scores were divided by 10 so that the odds ratios were interpretable as the relationship between a 10-point change in CAT-SS (on a 100-point scale) and the likelihood of a suicide-related out-come. Separate analyses were conducted for each outcome, with and without adjustment of suicide attempt in the past year.

This study was powered to estimate an AUC of 0.8 with a 95% confidence interval of plus or minus 5%. Assuming an event rate of 10%, n = 250 subjects at the 6-month follow-up were required. A total of n = 247 subjects completed the CAT-SS at the 6-month follow-up.

## Results

Demographic characteristics of the study sample at baseline are presented in Table 1. Mental health diagnoses (current) at baseline as determined by administration of the SCID-5 [12] included: Bipolar Disorder I and II (3.9%), Major Depressive Disorder (26.6%), Alcohol Use

**Table 1. Sample demographic characteristics and current mental health conditions.**

| | Baseline<br>n = 305* | Participants with at Least 1 Follow-up Visit<br>n = 265* |
|---|---|---|
| Age | 47.2 ± 12.6 | 47.4 ± 12.6 |
| | 47 (22–77) | 47 (22–77) |
| Male | 247 (81.0%) | 215 (81.1%) |
| Race | **n = 305** | **n = 265** |
| Caucasian/White | 221 (72.5%) | 192 (72.5%) |
| Black or African American | 50 (16.4%) | 42 (15.9%) |
| Native American/Alaskan Native | 4 (1.3%) | 4 (1.5%) |
| Asian or Pacific Islander | 3 (1.0%) | 3 (1.1%) |

(*Continued*)

**Table 1.** (Continued)

|  | Baseline<br>n = 305* | Participants with at Least 1 Follow-up Visit<br>n = 265* |
|---|---|---|
| Multiracial/Other | 27 (8.8%) | 24 (9.1%) |
| Ethnicity | **n = 305** | **n = 265** |
| Hispanic or Latino/a | 45 (14.8%) | 37 (14.0%) |
| Education | **n = 304** | **n = 264** |
| High school education | 47 (15.5%) | 35 (13.2%) |
| Some college, no degree | 83 (27.3%) | 72 (27.3%) |
| Associate's or Bachelor's degree | 112 (36.8%) | 99 (37.5%) |
| Graduate degree | 62 (20.4%) | 58 (22.0%) |
| Marital Status | **n = 305** | **n = 265** |
| Married | 131 (43%) | 118 (44.5%) |
| Single | 79 (25.9%) | 73 (27.6%) |
| Cohabitating | 15 (4.9%) | 12 (4.5%) |
| Widowed/Divorced/Separated | 80 (26.3%) | 62 (23.4%) |
| Sexual Orientation | **n = 305** | **n = 265** |
| Gay/Lesbian/Queer | 16 (5.2%) | 13 (4.9%) |
| Heterosexual | 280 (91.8%) | 244 (92.1%) |
| Bisexual | 9 (3%) | 8 (3.0%) |
| Employment Status | **n = 303** | **n = 263** |
| Employed Full-Time | 88 (29%) | 79 (30.0%) |
| Employed Part-Time | 31 (10.2%) | 29 (11.0%) |
| Unemployed, not currently seeking employment | 74 (24.4%) | 60 (22.8%) |
| Unemployed, seeking employment | 37 (12.2%) | 32 (12.2%) |
| Retired | 73 (24.1%) | 63 (24.0%) |
| Branch of Military Service | **n = 304** | **n = 265** |
| Army | 180 (59.2%) | 156 (58.8%) |
| Air Force | 46 (15.1%) | 41 (15.5%) |
| Navy | 34 (11.2%) | 31 (11.7%) |
| Marines | 32 (10.5%) | 25 (9.4%) |
| Multiple Branches | 12 (4%) | 12 (4.5%) |
|  | **n = 305** | **n = 265** |
| Number of Deployments | 2.1 ± 3.5 | 2.2 ± 3.7 |
|  | 1 (0–40) | 1 (0–40) |
| Number of Deployments to Combat Zone | 1.1 ± 1.8 | 1.2 ± 1.9 |
|  | 1 (0–20) | 1 (0–20) |
| Years of Military Service | 9.4 ± 7.5 | 9.4 ± 7.5 |
|  | 6.2 (0.5–39) | 6.4 (0.5–39) |
| **Current Mental Health Conditions** | **n = 305** | **n = 265** |
| Bipolar Disorders | 12 (3.9%) | 11 (4.2%) |
| Major Depressive Disorder | 81 (26.6%) | 69 (26.0%) |
| Alcohol Use Disorder | 27 (8.9%) | 25 (9.4%) |
| Substance Use Disorder | 28 (9.2%) | 24 (9.1%) |
| Generalized Anxiety Disorder | 11 (3.6%) | 10 (3.8%) |
| Sleep Disorders | 21 (6.9%) | 20 (7.6%) |
| Post-Traumatic Stress Disorder | 87 (28.5%) | 71 (26.8%) |

* n (%) or (Mean ± SD; Median (Range))

**Table 2. Columbia-Suicide Severity Rating Scale outcomes* at baseline, and 6 and 12 month visits.**

| | Baseline Visit n = 305 | | Participants with at Least One Follow-up Visit n = 265 | | 6 Month Visit n = 247 | 12 Month Visit n = 184 |
|---|---|---|---|---|---|---|
| Outcome | Lifetime n (%) | Past 3 Months n (%) | Lifetime n (%) | Past 3 Months n (%) | Since Last Assessment n (%) | Since Last Assessment n (%) |
| Active ideation with plan and intent | 91 (29.8%) | 14 (4.6%) | 76 (28.7%) | 13 (4.9%) | 5 (2.0%) | 4 (2.2%) |
| Attempt | 97 (31.8%)** | 19 (6.2%) | 82 (30.9%)*** | 16 (6.0%) | 3 (1.2%) | 4 (2.2%) |
| Interrupted, Aborted or self-interrupted attempt, or Preparatory acts or behaviors | 107 (35.1%) | 24 (7.9%) | 91 (34.3%) | 20 (7.5%) | 8 (3.2%) | 9 (4.9%) |
| Combination of the Above | 142 (46.6%) | 38 (12.5%) | 122 (46.0%) | 32 (12.1%) | 11 (4.5%) | 10 (5.4%) |

*Outcomes are not mutually exclusive

**n = 303

***n = 263

Disorder (8.9%), Substance Use Disorder (9.2%), Generalized Anxiety Disorders (3.6%), and Sleep Disorders (6.9%). Current PTSD was determined by responses to the CAPS-5 [13], with 28.5% of the sample meeting PTSD criteria (n = 87).

Administration of the CAT-SS resulted in a median administration time of 107 seconds with median administration of 11 items to meet a precision threshold less than 5.0 points on the 100 point scale. At baseline, using CAT-SS thresholds [7], 137 (51.6%) of the participants were categorized as being at low, 125 (47.3%) at moderate, and 3 (1.1%) at high risk. Per the baseline C-SSRS, 91 (29.8%) had lifetime active ideation with a plan and intent, and 97 (32.0%) had a lifetime attempt. Data from the C-SSRS across all three study visits (baseline, 6-month, 12-month) are presented in Table 2.

As a continuous measure the CAT-SS was strongly associated with suicide-related outcomes (active ideation with plan and intent; attempt; interrupted, aborted or self-interrupted attempt, or preparatory acts or behaviors; and all outcomes combined) over a 12-month period, with the strength of the associations increasing with repeated longitudinal assessments (see Table 3). Analyses were also conducted to study the added predictive accuracy of the CAT-SS for future suicidal events, above and beyond the predictive accuracy of a suicide attempt within the past year. Findings suggested large increases in AUC for all 4 outcomes (active ideation with plan and intent chi-square = 15.80, df = 1, p<0.0001; attempt chi-square = 5.78, df = 1, p = 0.02; interrupted, aborted or self-interrupted attempt or preparatory acts or behaviors chi-square = 17.92, df = 1, p<0.0001; and, all outcomes combined chi-square = 9.86, df = 1, p<0.002). The ROC curves for active ideation with plan and intent, and attempt for past year suicide attempt, CAT-SS, and past year suicide attempt and CAT-SS are displayed in Figs 1 and 2.

Longitudinal analysis of these data revealed that for every 10-point change in the CAT-SS score there was between a 50 and 77% increase in the likelihood of a suicidal event across the 4 outcomes, all of which were statistically significant, or a 5-fold to almost 8-fold increase over the range of the scale. Moreover, adjusting for suicide attempt in the past year, revealed similar strong associations between the CAT-SS and suicidal event outcomes ranging from 36 to 73% or 4-fold to 7-fold increase across the range of the scale (see Table 4).

## Discussion

To address the pressing public health problem of suicide, efforts must be aimed at validating measures that can be used to evaluate suicide risk in both primary and specialty care medical

**Table 3. Areas under the curve by Columbia-Suicide Severity Rating Scale outcomes for the Computerized Adaptive Test Suicide Scale.**

| | AUC 95% (Confidence Intervals) |
|---|---|
| **6 Months** | |
| Active ideation with plan and intent | 0.81 (0.74, 0.87) |
| Attempt | 0.65 (0.19, 0.99) |
| Interrupted, aborted or self-interrupted attempt, or Preparatory acts or behaviors | 0.72 (0.52, 0.91) |
| Combination of the above | 0.74 (0.60, 0.89) |
| **6–12 Months** | |
| Active ideation with plan and intent | 0.91 (0.77, 0.99) |
| Attempt | 0.72 (0.46, 0.98) |
| Interrupted, aborted or self-interrupted attempt, or Preparatory acts or behaviors | 0.82 (0.69, 0.96) |
| Combination of the above | 0.82 (0.70, 0.94) |
| **All 12 Months**[*] | |
| Active ideation with plan and intent | 0.84 (0.77, 0.91) |
| Attempt | 0.77 (0.58, 0.95) |
| Interrupted, aborted or self-interrupted attempt, or Preparatory acts or behaviors | 0.83 (0.74, 0.91) |
| Combination of the above | 0.81 (0.73, 0.89) |

[*]Individual with an event at either 6- or 12-months

settings. Ideally, such measures would be rapidly administered (e.g., self-report) via an electronic platform, and personalized to individual patients. Among Veterans seeking care at a VAMC, the CAT-SS assessed suicide risk severity with a median of 11 items in under two

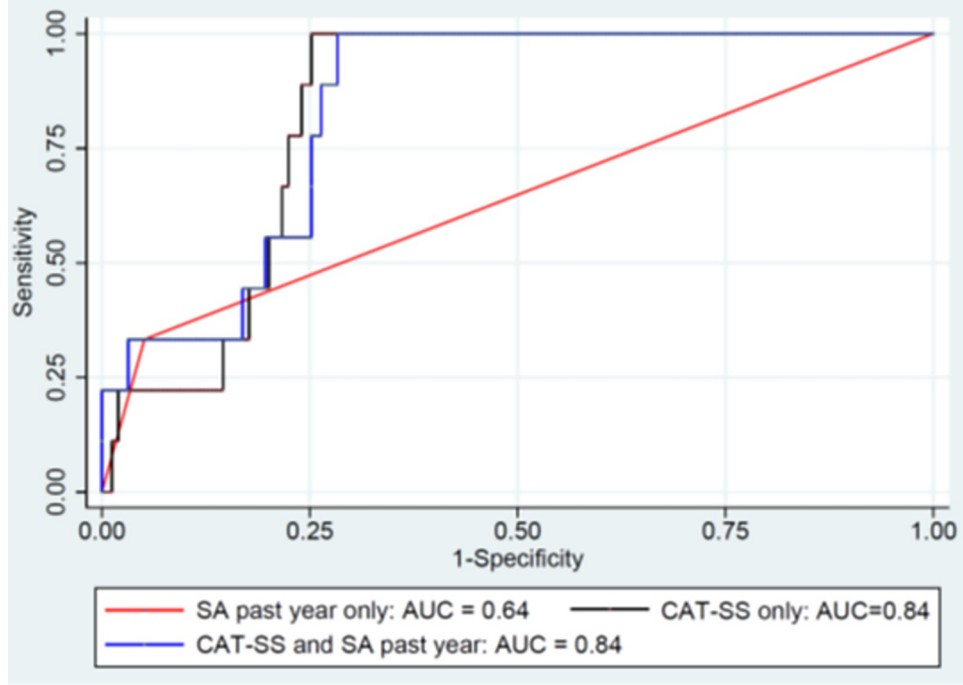

**Fig 1. Receiver operating characteristic curve for attempt in the past year and the Computerized Adaptive Test Suicide Scale, and both predicting active ideation with plan and intent over 12 months.**

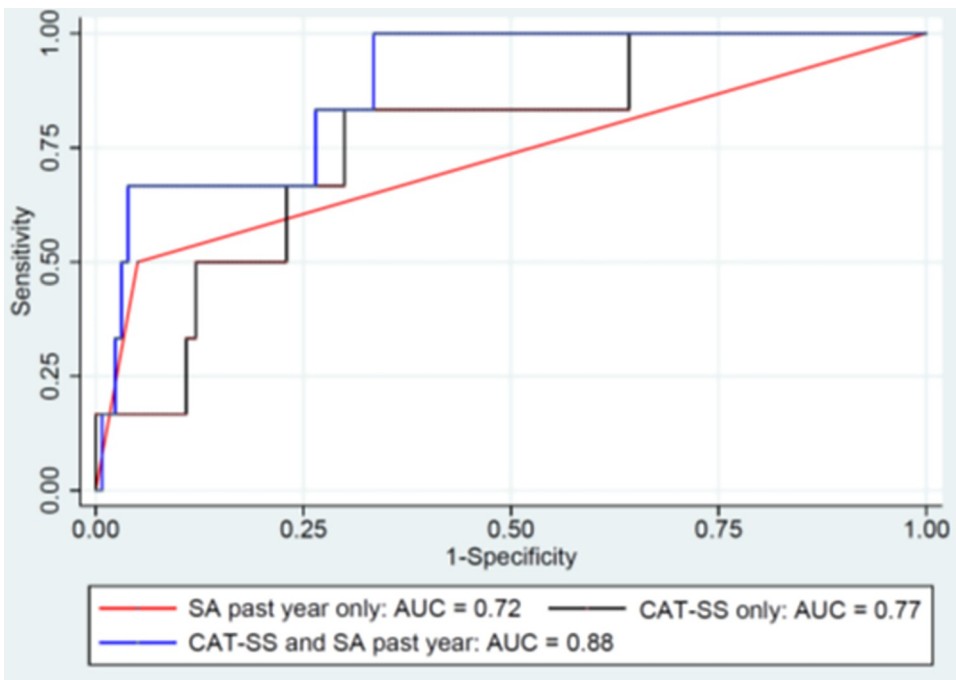

**Fig 2. Receiver operating characteristic curve for attempt in the past year and the Computerized Adaptive Test Suicide Scale, and both predicting actual attempt over 12 months.**

minutes (107 seconds); thereby highlighting feasibility of administration similar to that identified among the initial validation cohort (11 items and 110 seconds) [7].

Moreover, results revealed that CAT-SS scores were strongly associated with future suicide-related outcomes over the 12-month study period. Although results, in terms of such associations, were similar at 6- and 12-months, the strength of associations increased with repeated CAT-SS assessment. These findings highlight the utility of the CAT-SS for both initial identification and continued monitoring of risk. Longitudinal analysis also revealed that for every 10-point change in the CAT-SS score there was between a 50 and 77% increase in the likelihood of a suicidal event across the 4 outcomes, all of which were statistically significant, or a 5-fold to almost 8-fold increase over the range of the scale.

Previous research has shown that history of suicide attempt is one of the most significant risk factors for suicide [14]. Similarly, when clinicians were asked about factors which they considered "most important" in assessing suicide risk, they weighed the presence of suicide-related behaviors (e.g., preparatory behavior) as well as a history of attempts more heavily than other factors [15]. In fact, prior history of suicide attempt is strongly recommended as one of

**Table 4. Likelihood of a suicidal event per 10-point change in CAT-SS score.**

| Outcome | Odds Ratio (95% CI) *Unadjusted* | *p-value* | Odds Ratio (95% CI) *Adjusted for Past Year Suicide Attempt* | *p-value* |
|---|---|---|---|---|
| Active ideation with plan and intent | 1.50 (1.02, 2.19) | 0.04 | 1.36 (0.92, 2.03) | 0.13 |
| Attempt | 1.77 (1.23, 2.54) | 0.002 | 1.73 (1.14, 2.63) | 0.01 |
| Interrupted, aborted or self-interrupted attempt, or preparatory acts or behaviors | 1.74 (1.25, 2.41) | 0.0009 | 1.59 (1.14, 2.21) | 0.006 |
| Combination of the above | 1.63 (1.18, 2.25) | 0.003 | 1.55 (1.12, 2.15) | 0.009 |

the risk factors that should be assessed as part of a comprehensive suicide risk evaluation in the Departments of Veterans Affairs and Defense Clinical Practice Guideline for the Assessment and Management of Suicidal Behavior [14]. Thus, a critical marker of validity for any suicide risk measure is the degree to which it can predict future suicidal events when compared with other empirically robust variables, such as suicide attempt history. That is, the measure should increase the ability to predict future suicidal behavior, above and beyond known epidemiologic risk factors (e.g., history of a suicide attempt). In this study, CAT-SS scores outperformed history of suicide attempt in the past year as a predictor of future suicide-related thoughts and behaviors. As highlighted above, statistically significant increases in AUC were found in models that that added CAT-SS results to a model that only included a history of suicide attempt; thereby illustrating the added value of the CAT-SS over traditional predictive models based on past suicidal behavior only.

Recently, the CAT-SS has shown to be unbiased in a sample of 1,073 sexual and gender minority youth, mean age 20.3 years (SD = 3.2) [16], and to predict future suicidal events (ideation; plan; ideation, plan or attempt). Similar to our study, the CAT-SS improved predictive accuracy over traditional self-reports of ideation from an AUC of 0.70, 95% CI (0.64, 0.76) to AUC = 0.85, 95% CI (0.79, 0.90); suicide plan from AUC of 0.65, 95% CI (0.56, 0.73) to AUC = 0.84, 95% CI (0.77, 0.92); and, ideation, plan, or attempt from AUC = 0.71, 95% CI (0.65, 0.77) to AUC = 0.83, 95% CI (0.78, 0.88), all of which were statistically significant improvements in fit. The full model that included demographic characteristics, previous suicidal events, and the CAT-SS at baseline predicted suicidal ideation (AUC = 0.86, 95% CI (0.82, 0.91)), suicide plan (AUC = 0.86, 95% CI (0.80, 0.92)), and ideation, plan, or attempt (AUC = 0.84, 95% CI (0.79, 0.89)) at 6 month follow-up. Berona et al. [17] conducted a separate analysis of these data and showed predictive validity of the baseline CAT-SS in predicting time to suicide attempt during 6 months (HR = 1.34, 95% CI (1.03, 1.74)) overall and HR = 1.51, 95% CI (1.06, 2.15) for the transition from suicidal ideation to suicide attempt for each 10 point increment on the CAT-SS. These findings are remarkably similar to the findings of our study in a very different sample and age group, demonstrating the generalizability and robustness of our results.

These findings have important clinical implications for suicide risk screening across healthcare settings. The VHA has developed and implemented an enterprise-wide evidence-informed approach to suicide risk screening and evaluation, VA Suicide Risk Identification process (VA RISK ID) [5]. Currently, universal screening is being implemented using the C-SSRS Screener. However, findings from this study provide compelling evidence regarding both the efficiency and the long-term predictive validity of the CAT-SS in a medically diverse patient population. Further research is warranted to evaluate whether the CAT-SS could be feasibly implemented as part of universal screening efforts like the VA RISK ID [5], and whether CAT-SS dimensional scores could facilitate more accurate identification of suicide risk levels, while reducing patient and provider burden. Doing so, would be expected to provide additional time to facilitate personalized suicide risk-stratified care management.

As noted above, measures were administered as part of a research protocol, additional work is required to evaluate where the CAT-SS could be implemented in clinical settings. Efforts aimed at exploring this are warranted. Nonetheless, findings from this study suggest that if implemented in the electronic medical record, the CAT-SS would be expected to rapidly facilitate precise and personalized screening and assessment of suicide risk severity.

## Author Contributions

**Conceptualization:** Lisa A. Brenner, Lisa M. Betthauser, Nazanin Bahraini, Robert D. Gibbons.

**Data curation:** Lisa A. Brenner, Lisa M. Betthauser, Molly Penzenik, Robert D. Gibbons.

**Formal analysis:** Molly Penzenik.

**Funding acquisition:** Lisa A. Brenner.

**Investigation:** Lisa A. Brenner, Robert D. Gibbons.

**Methodology:** Nazanin Bahraini, Robert D. Gibbons.

**Project administration:** Lisa A. Brenner, Lisa M. Betthauser, Molly Penzenik.

**Resources:** Molly Penzenik.

**Supervision:** Lisa A. Brenner.

**Writing – original draft:** Lisa A. Brenner, Lisa M. Betthauser, Molly Penzenik, Nazanin Bahraini, Robert D. Gibbons.

**Writing – review & editing:** Lisa A. Brenner, Lisa M. Betthauser, Molly Penzenik, Nazanin Bahraini, Robert D. Gibbons.

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
