## [Decision Letter · Decision Letter 0]

23 Aug 2021

PONE-D-21-22855

Validation of a Computerized Adaptive Test Suicide Scale (CAT-SS) among United States Military Veterans

PLOS ONE

Dear Dr. Brenner,

Thank you for submitting your manuscript to PLOS ONE. After careful consideration, we feel that it has merit but does not fully meet PLOS ONE’s publication criteria as it currently stands. Therefore, we invite you to submit a revised version of the manuscript that addresses the points raised during the review process.

We look forward to receiving your revised manuscript.

Kind regards,

Sarah A. Arias

Academic Editor

PLOS ONE

Journal Requirements:

Reviewers' comments:

Reviewer's Responses to Questions

**Comments to the Author**

1. Is the manuscript technically sound, and do the data support the conclusions?

Reviewer #1: Yes

2. Has the statistical analysis been performed appropriately and rigorously? 

Reviewer #1: I Don't Know

3. Have the authors made all data underlying the findings in their manuscript fully available?

Reviewer #1: No

4. Is the manuscript presented in an intelligible fashion and written in standard English?

Reviewer #1: Yes

5. Review Comments to the Author

Reviewer #1: Thank you for an opportunity to review this interesting and well written paper reporting on Validation of a Computerized Adaptive Test Suicide Scale (CAT-SS) among United States Military Veterans. The study and the mss is a valid contribution to the current knowledge. Several point and queries for authors' consideration:

1. Introduction: Can the authors clarify "results from most rapid screeners like the PHQ-9 (6) identify non-negligible suicide risk versus risk severity or magnitude (7)" and “the full spectrum of suicidal symptomatology”? Can the authors provide more information regarding the 'computerized adaptive testing (CAT)' method?

2. Methods/participants: Can the authors provide more detailed information on the study recruitment procedure? Re 'n = 305; 6-month follow-up, n = 249; 12-month follow-up, n =185' - can the authors report on reasons for drop-out from the study?

3. Statistical analyses: Can authors clarify which instrument(s) allowed to measure 'active ideation with plan and intent; attempt; interrupted, aborted or self-interrupted attempt, or preparatory acts or behaviors'?

4. Discussion: Can authors clarify 'Thus, a critical marker of validity for any suicide risk measure is the degree to which it can predict future suicidal events when compared with other empirically robust variables, such as suicide attempt history'?

6. PLOS authors have the option to publish the peer review history of their article (what does this mean?). If published, this will include your full peer review and any attached files.

Reviewer #1: No

---

## [Author Response · Author response to Decision Letter 0]

2 Nov 2021

November 2, 2021

Members of the Editorial Board:

We are pleased to resubmit the following manuscript entitled: Validation of a Computerized Adaptive Test Suicide Scale (CAT-SS) among United States Military Veterans. Requests edits/actions are noted below (italics) and followed by responses.

1. It is important that you include a cover letter with your manuscript. Please ensure that this letter is addressed specifically to PLoS ONE. 

A cover letter that includes the necessary elements has been submitted.

2. It appears that your ORCiD iD has not been validated in your Editorial Manager account and we are unable to proceed until that step is complete. 

My ORCHID iD is : 0000-0002-2629-214X

I have attempted on a number of occasions to “fetch” my iD so that it can be entered into the system and received the following warning:

I have also sent an email to the editorial office regarding this matter. 

On 10/17/21, 5:09 PM, "Brenner, Lisa" <LISA.2.BRENNER@CUANSCHUTZ.EDU> wrote:

 Have resubmitted this manuscript - am getting error messages re: my orchid ID. Stating that it is associated with another account. Assistance would be super. 

3. Please include your full ethics statement in the 'Methods' section of your manuscript file. In your statement, please include the full name of the IRB or ethics committee who approved or waived your study, as well as whether or not you obtained informed written or verbal consent. If consent was waived for your study, please include this information in your statement as well.

The following has been added to the manuscript.

This study was conducted according to the guidelines laid down in the Declaration of Helsinki and all procedures involving human participants were approved by the Colorado Multiple Institutional Review Board (COMIRB).

and

Veterans were eligible if they were between the ages of 18 and 89 and able to provide written informed consent, which was obtained.

4. We note your updated Data Availability Statement: "Data cannot be shared publicly because of VA policies regarding data and security."

We have updated our Data Availability Statement. 

The full dataset cannot be made publicly available due to privacy concerns and restrictions imposed by the Colorado Multiple Institutional Review Board. All relevant de-identified data are included in the manuscript. For investigators with appropriate authorizations within the Department of Veterans Affairs, requests for data access can be made to VHAECHMIRECCAdmin@va.gov.

Many thanks for your assistance.

Lisa

Lisa A. Brenner, Ph.D.

Director, VHA Rocky Mountain MIRECC

Vice Chair of Research, Physical Medicine and Rehabilitation (PM&R)

Professor of PM&R, Psychiatry, & Neurology

University of Colorado, Anschutz Medical Campus 

Lisa.2.Brenner@cuanschutz.edu

303-724-4588

---

## [Decision Letter · Decision Letter 1]

14 Dec 2021

Validation of a Computerized Adaptive Test Suicide Scale (CAT-SS) among United States Military Veterans

PONE-D-21-22855R1

Dear Dr. Brenner,

We’re pleased to inform you that your manuscript has been judged scientifically suitable for publication and will be formally accepted for publication once it meets all outstanding technical requirements.

Kind regards,

Sarah A. Arias

Academic Editor

PLOS ONE

Reviewers' comments:

Reviewer's Responses to Questions

**Comments to the Author**

1. If the authors have adequately addressed your comments raised in a previous round of review and you feel that this manuscript is now acceptable for publication, you may indicate that here to bypass the “Comments to the Author” section, enter your conflict of interest statement in the “Confidential to Editor” section, and submit your "Accept" recommendation.

Reviewer #1: All comments have been addressed

2. Is the manuscript technically sound, and do the data support the conclusions?

Reviewer #1: Yes

3. Has the statistical analysis been performed appropriately and rigorously? 

Reviewer #1: Yes

4. Have the authors made all data underlying the findings in their manuscript fully available?

Reviewer #1: Yes

5. Is the manuscript presented in an intelligible fashion and written in standard English?

Reviewer #1: Yes

6. Review Comments to the Author

Reviewer #1: Thank you, the authors have addressed my queries and suggestions. More details as per above tick boxes.

7. PLOS authors have the option to publish the peer review history of their article (what does this mean?). If published, this will include your full peer review and any attached files.

Reviewer #1: No

---

## [Editor Report · Acceptance letter]

20 Dec 2021

PONE-D-21-22855R1 

Validation of a Computerized Adaptive Test Suicide Scale (CAT-SS) among United States Military Veterans 

Dear Dr. Brenner:

I'm pleased to inform you that your manuscript has been deemed suitable for publication in PLOS ONE. Congratulations! Your manuscript is now with our production department. 

Kind regards, 

on behalf of

Dr. Sarah A. Arias 

Academic Editor

PLOS ONE